# Recent Application of Nanomaterials to Overcome Technological Challenges of Microbial Electrolysis Cells

**DOI:** 10.3390/nano12081316

**Published:** 2022-04-12

**Authors:** Byeongcheol Kim, Euntae Yang, Bongkyu Kim, M. Obaid, Jae Kyung Jang, Kyu-Jung Chae

**Affiliations:** 1Technology Development Division, Korea Institute for Water Technology Certification (KIWATEC), 20 Gukgasandan-daero 40-gil, Guji-myeon, Dalseong-gun, Daegu 43008, Korea; kbc@kiwatec.or.kr; 2Department of Environmental Engineering, Kyungpook National University, 80 Daehak-ro, Buk-gu, Daegu 41566, Korea; 3Department of Marine Environmental Engineering, College of Marine Science, Gyeongsang National University, Tongyoung 53064, Korea; 4Division of Biotechnology, College of Environmental and Bioresource Sciences, Jeonbuk National University, Iksan 54596, Korea; bkim@jbnu.ac.kr; 5Chemical Engineering Department, Faculty of Engineering, Minia University, Al-Minya 61111, Egypt; mohammed.awad@kaust.edu.sa; 6Energy and Environmental Engineering Division, Department of Agricultural Engineering, National Institute of Agricultural Sciences, Rural Development Administration, 310 Nongsaengmyeong-ro, Deokjin-gu, Jeonju-si 54875, Korea; jkjang1052@korea.kr; 7Department of Environmental Engineering, College of Ocean Science and Engineering, Korea Maritime and Ocean University, 727 Taejong-ro, Yeongdo-gu, Busan 49112, Korea; ckjdream@kmou.ac.kr; 8Interdisciplinary Major of Ocean Renewable Energy Engineering, Korea Maritime and Ocean University, 727 Taejong-ro, Yeongdo-gu, Busan 49112, Korea

**Keywords:** hydrogen, microbial electrolysis cells, nanomaterials

## Abstract

Microbial electrolysis cells (MECs) have attracted significant interest as sustainable green hydrogen production devices because they utilize the environmentally friendly biocatalytic oxidation of organic wastes and electrochemical proton reduction with the support of relatively lower external power compared to that used by water electrolysis. However, the commercialization of MEC technology has stagnated owing to several critical technological challenges. Recently, many attempts have been made to utilize nanomaterials in MECs owing to the unique physicochemical properties of nanomaterials originating from their extremely small size (at least <100 nm in one dimension). The extraordinary properties of nanomaterials have provided great clues to overcome the technological hurdles in MECs. Nanomaterials are believed to play a crucial role in the commercialization of MECs. Thus, understanding the technological challenges of MECs, the characteristics of nanomaterials, and the employment of nanomaterials in MECs could be helpful in realizing commercial MEC technologies. Herein, the critical challenges that need to be addressed for MECs are highlighted, and then previous studies that used nanomaterials to overcome the technological difficulties of MECs are reviewed.

## 1. Introduction

Net zero emission, which refers to the balance between the emission and removal of greenhouse gases from human activity in the atmosphere, is now regarded as an essential goal for the survival of human beings and building a sustainable society [1]. To achieve net zero, innovations in energy production, which drastically reduce the use of fossil fuels and introduce carbon-neutral energy resources, are required [2]. Hydrogen is expected to play a vital role in energy innovation owing to its cleanliness and flexibility in production methods and applications. However, gray hydrogen, generated from natural gas and fossil fuels, accounts for approximately 95% of the current global hydrogen production. Unfortunately, a large amount of carbon dioxide is emitted during gray hydrogen production. The carbon footprint of gray hydrogen is estimated to be approximately 77.8 g CO_2_/MJ [3]. Therefore, green hydrogen production technologies need to substitute for fossil-fuel-based hydrogen production to make hydrogen fuels truly sustainable.

To date, several green hydrogen production technologies have been introduced, such as water electrolysis, dark fermentation, photofermentation, and microbial electrolysis cells (MECs) [4,5]. Among them, MECs have been attracting increased attention because this multidisciplinary device possesses merits over water electrolysis and other biological hydrogen production technologies due to the utilization of complementarily biological and electrochemical principles; in terms of energy requirement, MECs are more effective than water electrolysis [6]. Theoretically, MECs only need 0.14 V of applied voltage, while water electrolysis requires at least 1.2 V [7]. Moreover, MECs can harvest hydrogen gas from wastewater with higher purity and yield than other biological technologies, such as dark fermentation and photo fermentation [8].

Considering these merits, researchers have intensively studied MECs for over a decade for their commercialization [9]. According to our survey, using only the ‘microbial electrolysis cell’ keywords in the ‘web of science’ research database, 1428 research articles and 249 review articles have been published as of 6 January 2022. The number of studies pertaining to MECs has gradually increased over the years. Consequently, MEC systems have exhibited tangible advancements, and their technological readiness level has reached five [10,11]. Simultaneously, some critical technological and economic issues that need to be addressed, have been discovered in the implementation of competitive MECs for green hydrogen production. Recently, some studies have shown the potential of using nanomaterials with extraordinary properties as a good approach to overcome economic and technological challenges [12,13]. As nanotechnology advances rapidly, it is expected that various nanomaterials will be employed to improve MEC technologies; thus, it is important to identify the present status of nanomaterial applications in MECs.

This short review begins by introducing the principles of MECs and challenges in the commercialization of MECs. Then, the classification and synthesis approaches for nanomaterials are briefly reviewed. Finally, trials employing nanomaterials to enhance the performance and economic viability of MECs are discussed.

## 2. Microbial Electrolysis Cells

### 2.1. Principle of Microbial Electrolysis Cells

MECs are biocatalyst-assisted electrolysis devices for hydrogen generation [5]. A typical MEC reactor configuration is shown in Figure 1; a typical MEC reactor consists of two compartments, the anode and cathode chambers, separated by a physical separator. In the anode and cathode chambers, an anode electrode covered by electrochemically active bacteria (EAB) and a cathode electrode decorated with hydrogen evolution catalysts, respectively, are installed. During the operation of MEC, electrons and protons are released when the organic matter in wastewater is degraded into carbon dioxide by EAB in the anode chamber. The electrons released from EAB via transmembrane transfer flow into an anode electrode through three routes: (1) direct electron transfer if EAB directly adheres to an anode electrode, (2) electron transfer through conductive nanowires created by a specific EAB, and (3) electron shuttle by mediators. Subsequently, the electrons migrate into the cathode electrode. Simultaneously, the generated protons travel into the cathode chamber across a physical separator to ensure electrical neutrality. In the cathode chamber, the electrons and protons originating from the anodic oxidation of organic matter are utilized for hydrogen production on the cathode electrode with the assistance of electrocatalysts. However, this electrochemical redox reaction is non-spontaneous because of the thermodynamic barrier. Thus, to generate hydrogen, an energy of at least 0.14 V, corresponding to the theoretical value for overcoming the thermodynamic barrier, needs to be supplied from an external source [14].

### 2.2. Challenges of Microbial Electrolysis Cells

The commercialization of MECs for hydrogen production currently remains a stagnant state owing to significant challenges. Figure 2 shows the critical problems faced by MECs. To start with cell design-related challenges, a sure-fire breakthrough in cell design for scaling up is yet to be achieved. Although several scale-up designs have been suggested for MECs [15,16,17], they are unsuccessful because the strategies for scale-up MECs suffer a significant drop in hydrogen production efficiencies compared to that of lab-scale MECs [10]. For scaling up MEC systems, not only are the sizes of reactors and components enlarged, but the number of cell units and components utilized is also increased [18]. During the scale-up process of MECs, negligible problems in lab-scale systems became significant, or unexpected new issues appeared frequently. In a one-dimensional scale-up approach for an easily accessible method that increases the volume of the reactor, the use of a large electrode should lead to high energy losses owing to the electrical resistance of the electrode [19]. In addition, a large reactor with large electrodes would incur losses due to the mixed potential caused by the nonuniform electrode reaction of point-by-point of the whole electrode [20]. In addition, the junction resistance between the electrode and the current collector may cause a non-negligible loss. Therefore, modularizing unit cells of an appropriate size could be a feasible approach for cell design and strategies for scale-up of MECs.

Even if the stable performance of the entire system is secured through modularization, some issues still need to be addressed. To operate modularized MECs, it is necessary to secure sustainable energy sources as external power support for producing hydrogen in the cathode. At most lab scales, the MEC can be easily operated by applying a voltage based on the commercial power supply, but for the modularized MEC operation, multiple power supplies are required, which adversely affects the price competitiveness of the entire system. To solve these issues, two realistic strategies would be suggested. The first is the development of an electric circuit-device that can supply a constant voltage from one power supply to multiple cells. In the past, use of electric circuits was avoided in microbial electrochemical systems owing to the problem of energy loss. However, in recent years, their use has increased greatly as they stably provide high efficiency [21]. Another method is to employ alternative power sources based on renewable energy [22], such as solar cells and micro wind power; they can be applied to each unit cell separately. In addition, for the sustainable supply of power from renewable energy sources, a particular electric circuit that can provide a buffer (i.e., capacitor or battery) between the renewable energy source and the MEC circuit must be developed. Therefore, ultimately, we need to find a way to overcome this through convergence with electrical engineering, rather than MEC’s own technology.

Simultaneously, it is necessary to detect and improve measures for the stable operation of the MEC and minimize the losses that occur in the system in each main part, that is, the anode, cathode, and separator. Each component has its own ohmic resistance, including electrolyte resistance, and is affected by the charge transfer resistance and mass transfer during the operation of the system for the production of hydrogen [23]. However, the circumstances under which the specific resistance associated with each part affects the performance of MECs are not known. Therefore, the analysis method, which is currently not mature, should be improved to distinguish and classify the resistance of each part of the MEC and to minimize the overall loss of the entire system [24].

Apart from the electron losses of each component, there are still many issues related to electrodes and separators. In the anode, electron loss by methanogens is still an issue. In the cathode, the price competitiveness of catalysts must be secured, and highly stable catalysts and electrode materials for long-term operation need to be developed. Membrane-related issues, such as gas crossover, pH imbalance, biofouling, long-term stability, and price competitiveness, must be resolved. To seek clues for solving these problems, herein, we review the attempts that have been made to overcome the challenges of MEC based on the use of nanomaterials.

## 3. Nanomaterials Used in Microbial Electrolysis Cells

The development of nanomaterials has often provided a breakthrough to overcome technological issues in diverse engineering fields because nanomaterials have various distinctive and outstanding properties that are not present in macro-or bulk-sized counterparts. Owing to their preferred properties, such as high electrical conductivity, high specific surface area, high durability, high cost-effectiveness, good catalytic capability for hydrogen evolution, and anti-microbial activity, various nanomaterials have been developed and applied to mitigate some of the aforementioned challenges of MECs by improving the characteristics and performance of critical components (i.e., anode/cathode electrodes, catalysts, and separators) or substituting high-cost components, as shown in Figure 3 [12,25].

To synthesize nanomaterials applied for MECs, various approaches for nanomaterials, such as thermal annealing, electrochemical anodization, and electrodeposition, have been reported, as summarized in Table 1. The thermal annealing method was used to smoothen the metal by exposing it to a high temperature above the recrystallization temperature. Fan et al., (2011) synthesized Au and Pd NPs with various morphologies between 600 °C and 800 °C [26]. Electrochemical anodization was used for surface modification by immersing the workpiece in an aqueous electrolyte. Kim et al., (2018) synthesized an external TiO_2_ nanotube array photoanode by electrochemical anodization of Ti foil with an ethylene glycol electrolyte [27]. Electrodeposition is an in-situ metallic coating method that uses electrical current to reduce the cations of desired materials from an electrolyte. Several research groups have reported that electrodeposition increases the hydrogen production rate in MECs [28,29,30,31,32,33].

Jayabalan et al., (2020, 2021) synthesized nickel molybdate (NiMoO_4_), nickel oxide (NiO), and cobalt oxide (Co_3_O_4_) NPs as cathode catalysts [34,35]. Rani et al., (2021) also reported the synthesis of magnetite (Fe_3_O_4_) NPs to fabricate cathode catalysts [36] using a chemical precipitation method.

Electrospinning is used to fabricate polymeric nanoscale fibers with diameters in the sub-micrometer to nanometer range using a high-voltage power supply. Park et al., (2017) fabricated a sulfonated poly (arylene ether sulfone) (SPAES)/polyimide nanofiber (PIN) composite proton exchange membrane via electrospinning with an average fiber thickness of 200 nm [37]. In addition, various synthesis methods, such as solution plasma, hydrothermal synthesis, and chemical reduction, have also been adopted.

**Table 1 nanomaterials-12-01316-t001:** Summary of nanomaterials employed in microbial electrolysis cells.

Application	Nanomaterial	Structure	Size (nm)	Synthesis Method	Performance	Ref.
Anode electrode	Au nanoparticle	0D	0.33 μm^2^	Thermal annealing	Current density: 74.4 μA/cm^2^	[26]
Anode electrode	Pd nanoparticle	0D	0.35 μm^2^	Thermal annealing	Current density: 74.4 μA/cm^2^	[26]
Photo-anode electrode	TiO_2_ nanotubes	1D	Length: 4.04–4.35 μm	Anodization method	Current density: 0.371 mA/cm^2^ H_2_ production rate: 1434 mmol/m^3^/h	[27]
Photo-anode electrode	CeO_2_–rGO nanocomposite	2D	-	rGO nanosheets: modified Hummer’s method and thermal reduction; CeO_2_–rGO nanocomposite electrode: Polymerization and carbonization	H_2_ production rate: 5 m^3^/m^3^/d Cathodic H_2_ recovery efficiency: 95%	[38]
Cathodic catalyst	Pt nanoparticle	0D	7–20		H_2_ conversion efficiency: 80.6%	[39]
Cathodic catalyst	Ni nanoparticle	0D	7–20	-	H_2_ conversion efficiency: 73.0%	[39]
Cathodic catalyst	Pt–Ni nanoparticle (atomic ratio 1:1)	0D	7–20	-	H_2_ conversion efficiency: 76.8%	[39]
Cathodic catalyst	Pt–Cu nanoparticle (atomic ratio 1:1)	0D	7–20	-	H_2_ conversion efficiency: 72.6%	[39]
Cathodic catalyst	Ni nanoparticle	0D	30–50	Electrodeposition	H_2_ conversion efficiency: 82%	[28]
Catalyst	Ni nanoparticle	0D	40	Solution plasma	CH_4_ production enhancement: ~52.4%	[40]
Cathodic catalyst	Pd nanoparticle	0D	10–100	Bioelectochemical deposition	Cathodic H_2_ recovery efficiency: 65.5%	[41]
Cathodic catalyst	Ni_2_P nanoparticle	0D	7	Solution-phase method	Cathodic H_2_ recovery efficiency: 65.5%	[42]
Cathodic catalyst	Ni–Co–P nanoparticle	0D	33–35	Electrodeposition	H_2_ conversion efficiency: 90.3%	[29]
Cathodic photocatalyst	NiFe_2_O_4_ nanoparticle	0D	>17	Electrodeposition/spin coating	Current density: 0.74 A/m^2^ H_2_ production rate: 288 μmol/h/g	[30]
Cathodic catalyst	NiMoO_4_ nanoparticle	0D	<50	Sonochemical precipitation	H_2_ conversion efficiency: 11.96%	[34]
Cathodic catalyst	NiO nanoparticle	0D	-	Chemical precipitation	Cathodic H_2_ recovery efficiency: 27%	[35]
Cathodic catalyst	Co_3_O_4_ nanoparticle	0D	-	Chemical precipitation	Cathodic H_2_ recovery efficiency: 26%	[35]
Cathodic catalyst	Fe_3_O_4_ nanoparticle	0D	12–28	Chemical precipitation	Current density: 15.2 mA/m^2^	[36]
Cathodic catalyst	Carbon nanoparticle	0D	50	-	H_2_ conversion efficiency: 47%	[39]
Cathodic photocatalyst	TiO_2_ nanorod	1D	Length: 700 Diameter: 40	Hydrothermal method	H_2_ production rate: 4.4 μL/h	[43]
Cathodic photocatalyst	MoS_2_ nanosheet–TiO_2_ nanotube composite	1D	TiO_2_ nanotube diameter: about 100	Anodization method+ bioelectrochemical deposition	H_2_ production rate: 0.003 m^3^/m^3^/min	[44]
Cathodic catalyst	Mo_2_N nanobelt	1D	-	Hydrothermal synthesis + thermal annealing	H_2_ conversion efficiency: 74%	[45]
Cathodic catalyst	CoP nanoarray	1D	-	Hydrothermal synthesis + thermal annealing	H_2_ conversion efficiency: 34%	[46]
Cathode electrode	Polyaniline/MWCNT ^(1)^	1D	-	MWCNT: CVD ^(2)^; polyaniline deposition: in situ chemical oxidation polymerization	Cathodic H_2_ recovery efficiency: 42%	[47]
Cathode electrode	Polyaniline/MWCNT	1D	-	MWCNT: CVD; polyaniline deposition: in situ chemical oxidation polymerization	Cathodic H_2_ recovery efficiency: 56.7%	[48]
Cathode electrode	MoS_2_/CNT ^(3)^	1D	Outer/inner diameter: 7/3 Length: 10,000	CNT: CVD; MoS_2_ deposition: Hydrothermal method	Cathodic H_2_ recovery efficiency: 49%	[13]
Cathodic catalyst	SWCNT ^(4)^	1D	-	CVD	H_2_ conversion efficiency: 38.9%	[39]
Cathodic photocatalyst	Polyaniline nanofibers	1D	Thickness: 50	Oxidizing aniline at a perchloric acid/dichloromethane interface	H_2_ conversion efficiency: 79.2%	[49]
Supporting material for membrane mechanical strength reinforcement	Polyimide nanofiber	1D	Thickness: 200	Electrospinning	Membrane tensile strength: >40 MPa H_2_ conversion efficiency: 32.4%	[37]
Cathodic catalyst	Graphene	2D	-	GO nanosheets: Hummer’s method; graphene deposition: hydrothermal method	H_2_ production rate: 2.2 m^3^/m^3^/d	[50]
Cathodic catalyst	Mg(OH)_2_/graphene nanocomposite	2D	-	GO nanosheets: modified Hummer’s method; Mg(OH)_2_/graphene nanocomposites: Hydrothermal method	H_2_ conversion efficiency: 71% Cathodic H_2_ recovery efficiency: 83% H_2_ production rate: 0.63 m^3^/m^3^/d	[51]
Cathodic catalyst	NiO–rGO nanocomposte	2D	-	GO nanosheets: modified Hummer’s method; NiO–rGO nanocomposites: chemical reduction	H_2_ production rate: 4.38 mmol/L/d Cathodic H_2_ recovery efficiency: 21.2%	[52]
Cathodic catalyst	NiCo_2_O_4_–rGO nanocomposite	2D	-	GO nanosheets: modified Hummer’s method; NiO–rGO nanocomposites: chemical reduction	H_2_ production rate: 3.66 mmol/L/d Cathodic H_2_ recovery efficiency: 18.2%	[53]
Cathodic catalyst	MoS_2_ nanosheet	2D	150–250	Chemical exfoliation by Li intercalation	H_2_ production rate: 0.133 m^3^/m^3^/d Current density: 0.6 mA/cm^2^	[54]
Cathode electrode	MoS_2_/N-doped graphene nanocomposite	2D	-	GO nanosheets: modified Hummer’s method; MoS_2_ nanosheet: chemical exfoliation by Li reduction; MoS_2_–N–GO nanocomposites: hydrothermal method	H_2_ production rate: 0.19 m^3^/m^3^/d	[55]
Cathodic catalyst	MoS_2_–GO ^(^^5)^ nanocomposite	2D	-	Solvothermal method	H_2_ production rate: 0.183 m^3^/m^3^/d	[56]
Cathodic electrode	MoS_2_–Cu–rGO ^(6)^ nanocomposite	2D		GO nanosheets: modified Hummer’s method; MoS_2_–Cu–rGO nanocomposites: Hydrothermal method	H_2_ production rate: 0.449 m^3^/m^3^/d	[57]
Cathodic catalyst	MoS_x_ nanoparticle	3D	-	Electrodeposition method	Cathodic H_2_ recovery efficiency: 98%	[31]
Cathodic catalyst	Y Zeolites–NiO nanocomposite	3D	-	Y zeolites: Hydrothermal process; Y Zeolite–NiO nanocomposites: incipient wetness impregnation	H_2_ production rate: 0.83 m^3^/m^3^/d	[58]
Cathodic catalyst	NiO/MoO_2_/MoO_3_/C	3D		Electrodeposition method	Current density: 37.5 A/m^2^	[33]
Cathodic catalyst	CoNi/CoFe_2_O_4_ composite	3D	-	CoFe_2_O_4_: Hydrothermal method and calcination; CoNi/CoFe_2_O_4_ composite: Unpolar pulse electrodeposition	H_2_ production rate: 1.25 m^3^/m^3^/d	[33]
Cathodic catalyst	Activated carbon + Ni210 powder	3D	0.5–1 μm	-	H_2_ production rate: 0.28 m^3^/m^3^/d Cathodic H_2_ recovery efficiency: 98%	[59]
Antibiofouling membrane	Ag nanoparticle	3D	-	Chemical reduction	Biofouling reduction: 80.74% H_2_ conversion efficiency: 61.82%	[60]

^(1)^ MWCNT: multi-walled carbon nanotube; ^(2)^ CVD: chemical vapor deposition; ^(3)^ CNT: carbon nanotube; ^(4)^ SWCNT: single-walled carbon nanotube; ^(5)^ GO: graphene oxide; ^(6)^ rGO: reduced graphene oxide.

## 4. Nanomaterials Used in Microbial Electrolysis Cells

Table 1 summarizes the previous studies on nanomaterial applications in MECs. This section provides a detailed review of these previous studies, classified according to the application site and material type.

### 4.1. Anode Electrodes

To enhance the performance of anode electrodes and achieve sustainable photoassistance, 0D metal NPs were deposited onto the anode electrode to enhance the current densities in MECs [26]. Fan et al., (2011) deposited Au and Pd NPs of various shapes and sizes on graphite electrodes via thermal annealing and then evaluated the NP-decorated electrodes as anode electrodes in MECs. During MEC operation, the Au-decorated graphite anode electrode demonstrated up to 20 times higher current density generation than that of a plain graphite anode electrode, which implies that NP decoration can be one of the strategies to improve the performance of biocatalytic anode electrodes, although it may depend on the morphology, size, and composition of NPs.

In addition, Kim et al., (2018) fabricated a TiO_2_ nanotube array (Figure 4A) via an electrochemical anodization technique and installed it in the outer part next to a Pt-catalyzed cathode as a photoanode in a single-chamber MEC reactor [27]. During the operation of the MEC, the TiO_2_ nanotube photoanode was illuminated by a light source for the solar simulator. An MEC with a TiO_2_ photoanode under simulated solar light demonstrated a better performance (1434 mmol/m^3^/h of hydrogen production rate) compared to that under dark conditions, and the photoanode electrode can supply additional photoelectrons to facilitate the hydrogen evolution reaction at the cathode.

More recently, Pophali et al., (2020) used rGO nanosheets and CeO_2_ NPs to fabricate a novel photocatalytic carbon-based electrode (Figure 4B) [38]. Figure 4C shows a dope solution containing CeO_2_–rGO nanosheets that was cast and carbonized to fabricate the CeO_2_–rGO-incorporated carbon film electrode, and also shows the microscopic images of the electrode surface. According to this previous study, rGO nanosheets have a superior ability to accept electrons; when fabricating nanocomposites of CeO_2_ photocatalysts and rGO nanosheets, electron transfer from the conduction band of CeO_2_ to the other substances can be facilitated, which can decrease the probability of recombination of the electron–hole pairs. This led to the generation of more charged species, and consequently, the nanocomposite exhibited better photocatalytic activity. In their study, the CeO_2_–rGO-incorporated carbon film electrode was adopted as the photoanode. Under light irradiation, the CeO_2_–rGO nanocomposite carbon photoanode enhanced the hydrogen generation in MECs (cathodic hydrogen recovery of 98%).

### 4.2. Cathode Electrodes and Catalysts

#### 4.2.1. Metal-Based Nanomaterials

##### Metal Nanoparticles

Pt NPs are mostly used as cathodic proton-reduction catalysts because Pt is positioned near the top of the volcano plot, which is described using catalytic reaction and hydrogen adsorption [61]. However, owing to the high cost of Pt, some cost-effective nanomaterials substituting Pt have been employed as cathodic catalysts [29,30,39,40,42]. Many previous studies have used transition metal NPs (e.g., Ni and Cu) as cathodic catalysts in MECs. In a previous study conducted by Hrapovic et al., (2010), Ni NPs with relative size of 30–50 nm served as cathodic catalysts for hydrogen production in MECs. The Ni NPs were coated onto a carbon paper electrode via electrodeposition. The electrodeposited Ni NPs showed better catalytic performance than that of Pt. The electrodeposition of Ni NPs can significantly reduce the construction cost of MECs, and this method is suitable for manufacturing large-sized cathode electrodes of MECs because of its high reproducibility. Recently, Choi et al., (2019) evaluated Ni and Cu NPs, Pt and Ni NP mixtures (Pt–Ni), and Pt and Cu NP mixtures (Pt–Cu) as catalysts for hydrogen evolution reactions in MECs [39]. Small metal NPs (17–20 nm) were decorated on carbon NPs (CNPs), and the NP-decorated CNPs were immobilized on carbon felt electrodes using an air gun spray (Figure 5A). In their study, Ni and Cu NPs exhibited slightly lower catalytic efficiencies than that of Pt NPs. In the catalyst stability tests, the efficiencies of Cu and Ni for hydrogen production gradually decreased, whereas the efficiency of Pt was stable over time. In particular, Cu NPs exhibited a significant efficiency drop; correctly considering the price, stability, and catalytic capability, Ni NPs possess a high potential to substitute Pt in MECs for hydrogen production.

In addition, Ni NPs have been deposited on a cathode electrode (granular activated carbon (GAC)) in MECs to improve CH_4_ gas generation efficiency [40]. Notably, MEC systems can also produce CH_4_ instead of H_2_ using methanogenesis as a cathodic biocatalyst. The deposition of Ni NPs with an average size of 40 nm on GAC was performed using a solution plasma technique as follows: a pair of Ni electrodes was dipped into an aqueous solution containing GAC and exposed to plasma discharged by bipolar high-voltage pulses (Figure 5B). In their study, Ni NPs acted as a catalyst to accelerate the GAC-to-microorganism electron transfer.

Besides Ni, Pd NPs have been utilized as catalysts in MECs [41]. Wang et al., (2019) fabricated Pd NPs on a carbon cloth electrode using a novel deposition strategy of bioelectrochemical deposition. During bioelectrochemical deposition, *S*. *oneidensis* MR-1 microbes reduced Pd (II) to Pd NPs on the carbon cloth. During MEC with bioelectrochemically deposited Pd NPs, a significantly large amount of H_2_ was generated because smaller-size and higher-specific-area Pd NPs can be obtained using bioelectrochemical deposition. However, the Pd catalyst lacked a binding force to the electrode, and consequently, Pd NPs were lost.

##### Metal Compound Nanoparticles

Diverse transition metal oxides and NPs of other transition metal compounds have been explored as cathodic catalysts for hydrogen production in MECs. Kim et al., (2019) evaluated the hydrogen evolution reaction catalytic capability of Ni_2_P NPs with carbon block particles (Vulcan XC-72R) in MECs. Small nickel(II) phosphate (Ni_2_P) NPs with an average size of 7 nm were coated on carbon black using a solution-phase method (Figure 6A). The MEC equipped with Ni_2_P catalysts exhibited a comparable hydrogen production rate (0.29 L-H_2_/L-d) to MECs equipped with Pt and Ni catalysts. In another study conducted by Chaurasia et al., (2020), Ni–CO–P nanocomposite catalysts were prepared by electroplating on stainless steel 316 and copper rods, which were used as cathode electrodes in MECs. The Ni–CO–P catalyst exhibited superior stability against corrosion and enhanced hydrogen production reaction catalytic capability in MECs compared to those of the pristine cathodes. Tahir (2019) [30] used Ni-compound NPs to fabricate a photocatalytic cathode electrode. The NiFe_2_O_4_ NPs were deposited on a nanostructured tungsten trioxide (WO_3_)–fluorine tin oxide (FTO) glass electrode by electrodeposition or spin coating and employed as photocathodes in MECs. An MEC with a photo-cathode incorporated with 1.5 wt% nickel ferrite (NiFe_2_O_4_) NPs achieved the highest current density and hydrogen evolution rate under visible light irradiation owing to the highest Brunauer–Emmett–Teller (BET) surface area with the smallest crystalline size of 17 nm. In 2021, another Ni compound NP, nanocrystalline nickel molybdate (NiMoO_4_), was also used as a cathodic catalyst to produce hydrogen in MECs [10]. The NiMoO_4_ NPs were synthesized by sonochemical precipitation using nickel(II) nitrate hexahydrate (Ni(NO_3_)_2_·6H_2_O) and sodium molybdate(VI) dihydrate (Na_2_MoO_4_·2H_2_O) as the Ni and molybdenum (Mo) precursors, respectively. Figure 6B shows a microscopic image of the synthesized NiMoO_4_ NPs with a relative size of <50 nm. The prepared NiMoO_4_ NPs were coated on the Ni foam using a polyvinyl alcohol binder. In MEC tests fed with sugar industry effluent, the NiMoO_4_-NP-decorated Ni foam cathode showed a better hydrogen production performance (i.e., 0.12 L H_2_/L/d of hydrogen production rate and 11.96% overall hydrogen efficiency), compared to that of a pristine Ni foam.

As the cases for applying 0D transition metal oxide NPs to MECs, nickel(II) oxide (NiO) and cobalt tetraoxide (Co_3_O_4_) NPs have been assessed as cathode catalysts of sugar-industry-effluent-fed MECs [34]. The NiO and Co_3_O_4_ NPs were prepared using nickel(II) chloride (NiCl_2_) and cobalt(II) chloride (CoCl_2_), respectively, by a chemical precipitation method. The synthesized oxide NPs were deposited on Ni foams. The Ni foam electrodes deposited with NiO and Co_3_O_4_ generated more hydrogen than bare Ni foam in MECs. In addition, Rani et al., (2021) [36] used Fe_3_O_4_ NPs as cathodic catalysts in MECs. Iron oxide (Fe_3_O_4_) NPs with size range of 12–28 nm, synthesized by a co-precipitation method, were doped onto a graphite sheet and carbon cloth via a drop casting method. An MEC equipped with a Fe_3_O_4_ NP-doped carbon cloth electrode demonstrated 1.5 mA/m^2^ of current density.

Some metal/metal oxide-based nanocomposites in 3D structures have been employed for the development of hydrogen evolution reaction catalysts and antibiofouling membranes in MECs [31,58]. In the case of employing transition metal-based 3D nanomaterials as catalysts, Kokko (2017) electrochemically deposited MoSx on carbon electrodes for hydrogen production in an acidic catholyte in MECs [31]. In addition, to further reduce the overpotential of the MoSx cathode electrodes, they were heat-treated. Consequently, a high cathodic H_2_ recovery of approximately 90% was achieved in an MEC with a MoSx-deposited carbon electrode. Electrodeposition and impregnation followed by heat treatment resulted in the lowest overpotentials of the MoSx electrodes, and MoSx electrodes prepared using these methods demonstrated high cathodic hydrogen recoveries (approximately 90%) and H_2_ production rates (0.26–0.39 m^3^/m^3^/d). Thus, MoSx–carbon electrodes are a viable option for hydrogen evolution in MECs under practical conditions.

Wang et al., (2019) employed NiO-loaded Y zeolite (NiO/Y zeolite) nanomaterials as cathodic catalysts for hydrogen evolution in MECs and evaluated their catalytic ability by comparing them with Pt catalysts (Figure 6C) [58]. In their study, Y-zeolite was synthesized using sodium hydroxide, sodium aluminate, and amorphous silica via a hydrothermal technique. Subsequently, NiO/Y zeolite composites were obtained using the capillary impregnation technique. During the electrochemical test, the NiO/Y zeolite composites exhibited the highest activity among materials for the hydrogen evolution reaction. During the operation of MECs, a slightly higher hydrogen production rate of 0.83 m^3^/m^3^/d was observed with the NiO/Y zeolite catalysts than that with Pt catalysts. The synergistic effect of the good catalytic activity of NiO and the high surface area and porous structure of Y zeolites leads to excellent hydrogen production in MECs.

Zhao et al., (2019) also employed transition metal oxide nanocomposites, such as NiO/MoO_2_/MoO_3_/C, for hydrogen-producing MECs [32]. Metal oxide nanocomposites were directly synthesized on a carbon paper electrode via electrodeposition. The NiO/MoO_2_/MoO_3_/C nanocrystalline layer exhibited a laminar structure in microscopic observation (Figure 6D). The NiO/MoO_2_/MoO_3_/C-deposited electrode was installed as the cathode electrode in an MEC. According to their study, the hybrid laminar-structured nanocrystalline catalysts not only exhibited higher hydrogen production than Pt but also demonstrated robust durability.

More recently, Fang et al., (2021) fabricated a new cathode electrode by employing cost-effective catalysts with competitive catalytic abilities, Co and Ni, and highly electrically conductive Fe [33]. The combination of these transition metal-based materials possibly improves the electron transport and electrical conductivity of electrodes. They grew 3D-structured CoNi/CoFe_2_O_4_ nanocomposite catalyst on a Ni foam electrode via a two-step strategy: (1) hydrothermal step followed by (2) unipolar pulse electrodeposition step, as illustrated in Figure 6E. The CoNi/CoFe_2_O_4_ catalysts possess a durian-like structure (Figure 6E); durian-like 3D nanomaterials have been tested as catalysts for hydrogen production in MECs. During MEC test, CoNi/CoFe_2_O_4_ catalysts not only exhibited an excellent hydrogen production rate (1.25 m^3^/m^3^/d) but also good stability. The unique 3D structure derived from the combination of CoFe_2_O_4_ and CoNi double-layered hydroxide nanosheets and Ni foam could yield this good performance by providing large catalytic active sites, enhanced mass transport, and a large specific surface area.

##### Transition Metal Nanobelt and Nanotube

The input of an external voltage above a certain level is required to overcome the thermodynamic barrier and generate hydrogen in MECs. This is considered to be one of the critical issues faced by MECs, impeding their sustainability. Therefore, some studies have suggested the use of solar energy to sustainably overcome the thermodynamic barrier. Moreover, to enhance the hydrogen production efficiency, researchers have attempted to harness solar energy in MECs. Titanium dioxide (TiO_2_) is one of the most widely used photocatalytic materials owing to its low cost, non-toxicity, and stability. Critical factors related to the photocatalytic ability of TiO_2_ include its crystallinity, size, and surface area [43]. The 1D TiO_2_-based nanomaterials possess significant merits in terms of these factors over other TiO_2_ nanomaterials owing to their easily tunable morphological properties, such as length and thickness [43]. Therefore, in many previous studies, TiO_2_-based 1D nanomaterials have been adapted to transform solar energy into a built-in bias that facilitates hydrogen production, owing to their excellent photocatalytic abilities [27,43,44].

In a previous study conducted by Chen et al., (2013) [43], TiO_2_ nanorods were first employed as a photo-cathode electrode, which can respond to light irradiation in MECs. The TiO_2_ nanorods were hydrothermally synthesized, and the TiO_2_ nanorod arrays are shown in Figure 7A. An MEC equipped with TiO_2_ nanorod electrodes successfully generated hydrogen at a rate of 4.4 μL/h without an applied voltage.

Zeng et al., (2019) recently constructed MoS_2_-doped TiO_2_ nanotube electrodes for photodriven MECs [44]. To improve the photocatalytic activity of the TiO_2_ nanotubes prepared via anodization, polydopamine (PDA) was initially coated on the TiO_2_ nanotubes. Molybdenum disulfide (MoS_2_) nanomaterials were synthesized in situ on PDA-coated TiO_2_ nanotubes installed in an MEC cathode chamber filled with ammonium tetrathiomolybdate ((NH_4_)_2_[MoS_4_]) solution during the operation of the MEC under visible light illumination. The performance of the prepared MoS_2_–PDA–TiO_2_ cathode electrodes was tested in MECs under visible-light illumination. In the absence of external power input, the MEC equipped with the MoS_2_–PDA–TiO_2_ electrodes successfully generated hydrogen by the mechanism involved in the generation of electrons by anodic biocatalysts and capturing electrons by photo-excited holes on the photocathodes.

A CoP nanoarray was created on the surface of a Ni foam electrode for hydrogen production in MECs [46]. The CoP nanoarray was synthesized on the Ni foam electrode via a hydrothermal step to deposit Co precursors and a thermal annealing step to deposit phosphate Co precursors on the Ni foam electrode. Microscopic images of the CoP nanoarrays are shown in Figure 7B. During the hydrogen production performance evaluation of catalysts in MECs, the CoP nanoarray-coated Ni foam electrode demonstrated enhanced hydrogen production efficiency compared to those of bare and Pt-coated Ni foam.

#### 4.2.2. Carbon-Based Nanomaterials

##### Carbon Nanoparticles

Choi et al., (2019) [39] used carbon NPs (or carbon black) to fabricate cathode electrodes (Figure 5A). Owing to their high surface area and good electrical conductivity, carbon NPs were selected as the cathodic catalysts. However, plain carbon NPs exhibit poorer catalytic capability than that of metal NP-modified carbon NPs. An MEC with a cathode coated with pristine carbon NPs only achieved 47% hydrogen conversion efficiency. These carbon NPs could serve to increase the surface area and electrical conductivity of cathode electrodes, including high-performance metal catalysts such as Ni and Pt NPs.

##### Carbon Nanotubes

Single-walled carbon nanotubes (SWCNTs) and multi-walled carbon nanotubes (MWCNTs) have also been adopted as cathode electrode materials to replace expensive Pt catalysts in hydrogen-producing MECs because of their outstanding properties, such as high surface area, high electrical conductivity, and ease of functionalization [39,47,48,49]. However, pure CNTs did not exhibit comparable or higher hydrogen production performance than Pt; for example, Wang et al., (2012) compared the hydrogen production of pristine MWCNTs, Pt, and Pt-coated MWCNTs in MECs [62]. An MEC with a pristine MWCNT catalyst demonstrated a much lower hydrogen production rate than those with Pt-coated MWCNTs and Pt catalysts. Similarly, Choi et al., (2019) demonstrated that an MEC equipped with an SWCNT-coated carbon felt electrode produced hydrogen at a lower rate than that of MECs equipped with transition metal NP-coated electrodes [39]. Therefore, to improve hydrogen evolution with CNTs, CNTs need to be modified with other materials that possess high catalytic activities or are able to create synergistic effects with CNTs. Polyaniline has been used to modify MWCNTs owing to its attractive characteristics for electrodes, such as tunable electrical conductivity, excellent stability, and facile synthesis [47,48]. The polyaniline modification was conducted by adding MWCNTs to a hydrochloric acid (HCl) solution containing aniline monomers and dropping ammonium persulfate into the MWCNT dispersion. The use of polyaniline can not only enhance the hydrogen production capability of CNTs, but it can also make CNTs more hydrophilic. This facilitates cathode electrode fabrication using CNTs by homogeneously dispersing CNTs in the solution during cathode electrode fabrication. The polyaniline modified-CNT electrodes demonstrated an improved catalytic activity, increased electrochemical surface area, and cost-effectiveness; hydrogen was produced cost-effectively using an MEC with polyaniline–CNT electrode (cathodic hydrogen recovery: 42%; hydrogen production rate: 0.67 m^3^/m^3^/d) [47].

##### Graphene Derivatives

Many previous studies have demonstrated that graphene derivatives such as nitrogen-doped graphene (N–G), graphene oxide (GO), and reduced GO (rGO) are excellent building blocks for the development of cost-effective and high-performance cathodic catalysts for hydrogen production [38,52,53,55,56]. Cai et al., (2016) also used graphene to enhance the catalytic activity of Ni-foam electrodes [49]. A graphene-coated Ni foam cathode was prepared via a hydrothermal process (Figure 8A). The electrochemical performance of the Ni foam electrodes was improved by coating with graphene. An MEC with graphene-modified Ni foam exhibited a significantly improved hydrogen production rate (1.31 L H_2_/L/d) compared to that with a pristine foam, and a similar performance to that of Pt-catalyzed electrode.

Dai et al., (2016) also developed nano-Mg(OH)_2_/GO nanocomposites via a hydrothermal process using graphene oxide (GO) nanosheets and MgSO_4_·7H_2_O as precursors and hydrazine hydrate as additive (Figure 8B) [30]. In the performance evaluation of MECs, the nano-Mg(OH)_2_/GO nanocomposite-coated carbon paper electrode exhibited competitive hydrogen production performance compared to that of a Pt-coated electrode. The hydrogen conversion efficiency of the MEC using the nano-Mg(OH)_2_/GO nanocomposite electrode reached approximately 71%. In addition, the nano-Mg(OH)_2_/GO electrode demonstrated robust stability with a reasonable cost.

In addition, Jayabalan et al., (2020) prepared two types of metal oxide–rGO nanocomposites, NiO–rGO and Co_3_O_4_–rGO, as cathodic catalysts for hydrogen production in sugar industry wastewater-fed MECs [52]. In their study, GO nanosheets were synthesized using a modified Hummer’s method. The NiO–rGO and Co_3_O_4_–rGO nanocomposites were then synthesized and coated on Ni foam via wet chemistry using Ni and Co precursors (Figure 8C). The MEC equipped with NiO–rGO and Co_3_O_4_–rGO coated Ni foam cathode electrodes outperformed those equipped with uncoated Ni foam in terms of hydrogen production. Similarly, Jayabalan et al., (2020) [53] synthesized NiCo_2_O_4_–rGO nanocomposites and employed them as catalysts for MECs. The maximum hydrogen production rate achieved in the MEC with NiCo_2_O_4_–rGO catalysts was 0.14 m^3^/m^3^/d.

MoS_2_-based nanomaterials are a promising alternative cathode catalyst to Pt for hydrogen generation because of their comparable exchange current density and hydrogen adsorption ability to Pt. In previous research by Rozenfeld et al., (2018) [54], MoS_2_ nanosheets were used as cathodic catalysts for hydrogen production. The MoS_2_ nanosheets were chemically exfoliated from the parent MoS_2_ particles via lithium intercalation. The lateral size of the obtained MoS_2_ nanosheets was approximately 200 nm. These MoS_2_ nanosheets were much smaller than those of the pristine MoS_2_ catalysts. When the small MoS_2_ nanosheets were employed as cathodic catalysts, higher hydrogen production rates (0.133 m^3^/m^3^/d) were observed in the MEC compared to those with pristine MoS_2_ and Pt catalysts.

However, MoS_2_-based nanomaterials have several limitations as catalysts for hydrogen evolution reaction, such as low electrical conductivity and insufficient catalytically active sites for HER. To overcome the limitations of MoS_2_, 2D carbonaceous nanomaterials, such as GO and rGO nanosheets, have been employed for constructing MEC cathode electrodes with MoS_2_ owing to their outstanding physicochemical properties including, high aspect ratio, conductivity, specific surface area, and easy functionalization [55]. Hou et al., (2014) reported the synthesis of MoS_2_ nanosheet/N–G nanosheet composite cathode catalysts by a hydrothermal method [55]. The N–G nanosheets have various merits, such as good chemical stability, excellent electrical conductivity, and modest catalytic capability for hydrogen evolution. Moreover, N–G nanosheets offer an increased contact area for effective charge transfer and reduce the time and distance of the charge transport. This synergistic effect can enhance the catalytic activity. In their study, MoS_2_ and GO nanosheets were initially prepared by chemical exfoliation and a modified Hummer’s method. Subsequently, using a mixture of the synthesized MoS_2_ and GO nanosheets and ammonia, 3D MoS_2_/N–G aerogels were fabricated via a hydrothermal process (Figure 8D). An MEC with a MoS_2_/N–G aerogel cathode demonstrated an outstanding hydrogen production rate of 0.19 m^3^/m^3^/d with an applied voltage of 0.8 V, which was comparable to that obtained with a Pt catalytic electrode. This excellent performance is mainly attributed to its morphological characteristics (i.e., porous structure and 3D networks) and the synergistic effect of the co-catalysts.

Similarly, Hou et al., (2021) fabricated a MoS_2_–GO nanocomposite and loaded it in situ on Ni foam by a solvothermal process [56]. During the operation, the MEC with the MoS_2_–GO NF foam cathode showed an excellent hydrogen production rate of 0.183 m^3^/m^3^/d.

Dai et al., (2021) synthesized MoS_2_–Cu–rGO composites by a hydrothermal technique [57]. From the microscopic observation of the synthesized MoS_2_–Cu–rGO composites, it was found that MoS_2_ sheets were created vertically on the rGO surface because Cu_2_O served as the bridged absorbent and as the channel for effective charge transfer between MoS_2_ and rGO (Figure 8E). This morphological property can provide more active sites for hydrogen evolution as well as enhanced electrical conductivity. Thus, in their study, the MoS_2_–Cu–rGO composites exhibited a superior hydrogen production rate of 0.449 m^3^ H_2_/m^3^/d compared to that of the Pt/C cathode in MECs. Moreover, The MoS_2_–Cu–rGO cathode exhibited advantages in terms of cost and stability.

##### Other Carbon-Based Nanomaterials

Carbon-based 3D nanomaterials, such as activated carbon, have been used to increase the electrical conductivity and specific surface area of electrodes in MECs [59]. As stated earlier, Ni catalysts are one of the most reasonable alternatives to replace expensive Pt catalysts in MECs because of their comparable HER activity to that of Pt, and moderate cost. The capital cost of MECs can be reduced by using Ni catalysts as cathodes in MECs. To render MECs a more competitive technology for green hydrogen production, their capital cost should be reduced further. Reducing the quantity of Ni catalysts used to fabricate a cathode electrode can be a good strategy to further reduce the costs without deteriorating the performance of the cathode. Kim and Logan (2019) blended activated carbon with Ni powder to decrease the amount of Ni used for catalysts. This strategy also enhanced the catalytic activity by significantly increasing the active sites available for the HER [58]. A Ni powder-activated carbon blend-loaded cathode electrode (4.8 mg/cm^2^ of Ni power loading) successfully generated more hydrogen at a higher production rate of 0.38 L-H_2_/L/d than an only-Ni-powder-loaded cathode electrode (0.28 L-H_2_/L/d), even though 16 times less amount of Ni powder was used to prepare the Ni powder-activated carbon blend-loaded cathode electrode. In addition, the cathodic hydrogen recovery for the Ni powder-activated carbon blend-loaded cathode electrode reached 98%.

#### 4.2.3. Polymer Nanofiber

Polymer-based 1D nanomaterials, such as polymer nanofibers, have also been used to fabricate ion-exchange membranes and cathode photocatalysts. Jeon and Kim (2016) [49] designed a photocatalytic cathode electrode by depositing *p*-type polyaniline nanofibers on carbon cloth. The *p*-type semiconductor polyaniline nanofibers (band gap = 2.44 eV) used in their study were synthesized by oxidizing aniline at the interface between perchloric acid and dichloromethane. A photo-assisted MEC was constructed by installing a polyaniline nanofiber-coated carbon cloth electrode. The photo-assisted MEC stably produced hydrogen over six months with the assistance of external power and visible-light illumination. The hydrogen conversion efficiency of the photoassisted MECs with polyaniline photocatalysts was 79.2%.

As another case for application of polymer nanofiber in MECs, Park et al., (2017) [37] used electrospun polyimide nanofiber (PIN) web to fabricate mechanically reinforced sulfonated poly(arylene ether sulfone) (SPAES)-based proton exchange membranes. The polyimide nanofiber web effectively enhanced the mechanical strength of the membranes. In addition, the composite membranes exhibited excellent capability for selective proton transport, and ameliorated hydrogen production was achieved in MECs with the SPAES/PIN proton exchange membrane.

### 4.3. Membrane Modification

In addition to developing catalysts, 3D metal NPs have been adopted as anti-biofouling agents for fabricating membranes with high fouling resistance. Biofouling of a proton exchange membrane installed between the anode and cathode compartments in a two-chamber MEC is one of the most critical issues because it can hinder proton transport across the membrane. To retard biofouling, Park et al., (2021) modified proton exchange membranes by coating them with antibacterial Ag NPs [60]. In addition, Park et al., (2021) attempted to resolve the adverse effects of Ag NP coating on membranes, including Ag release and proton transport interference, by changing the coating method, such as coating only Ag NPs with ascorbic acid, coating with a polydopamine layer followed by Ag NP coating, and coating Ag NPs followed by coating with a polydopamine layer, as illustrated in Figure 9. According to their study, coating polydopamine and Ag NPs yielded a significantly higher hydrogen conversion efficiency (68.12%) over 6-month operation of MECs with 80.74% reduction in biofouling (compared to a pristine membrane) and non-sacrificed proton transportability (t^−^+) of 0.96. Additionally, in a coating procedure involving the coating of a polydopamine layer followed by Ag NPs, Ag NPs were more homogeneously formed, and less Ag was released.

## 5. Conclusions

In this review, various applications of different types of nanomaterials in MECs are reviewed. Many previous studies have shown that the use of nanomaterials can improve or mitigate and resolve several challenges of MECs, such as high material cost, performance and durability of electrodes, and membrane deterioration by biofouling. In particular, some of the developed nanomaterials have demonstrated the feasibility as alternative cost-effective and durable cathode materials for scaling up MECs. Moreover, with the assistance of renewable solar power, the application of some photocatalytic nanomaterials could help enhance the sustainability of MECs by eliminating the need for external power input. At present, the application potential of these nanomaterials in scale-up systems at practical sites still needs to be evaluated. However, nanomaterials can provide vital inputs for the commercialization of MECs if continuous efforts are made to optimize and upgrade nanomaterials.

## Figures and Tables

**Figure 1 nanomaterials-12-01316-f001:**
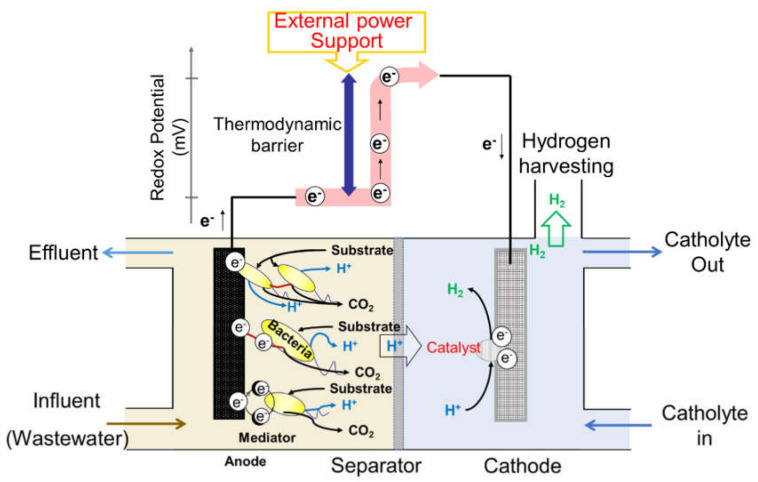
Schematic illustration of the working principle of microbial electrolysis cells.

**Figure 2 nanomaterials-12-01316-f002:**
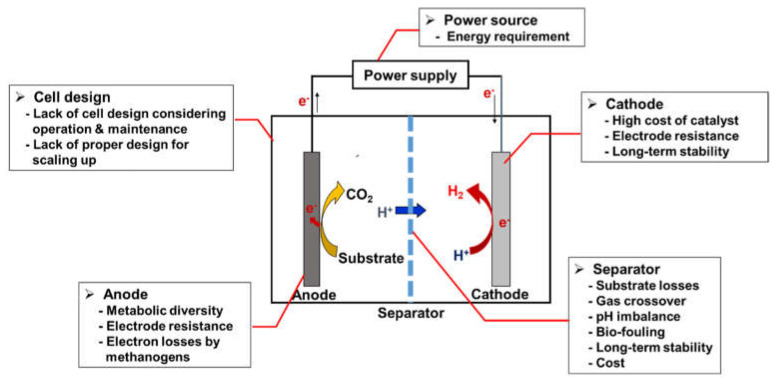
Major challenges of microbial electrolysis cells.

**Figure 3 nanomaterials-12-01316-f003:**
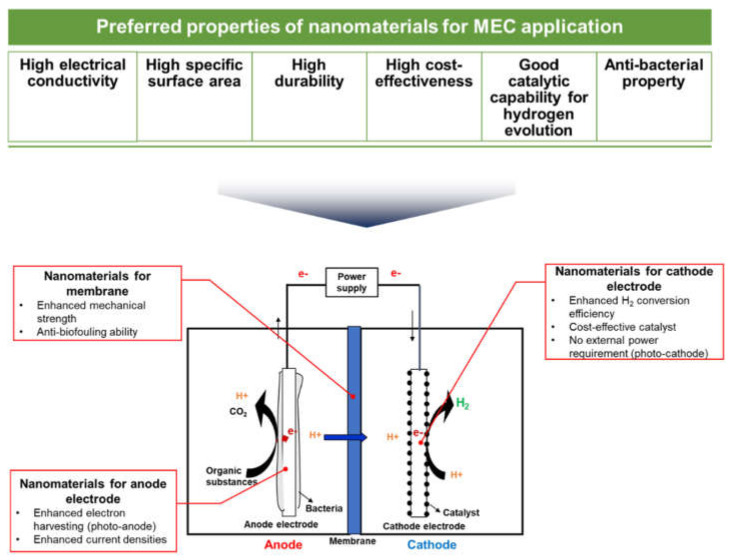
Various applications of nanomaterials in microbial electrolysis cells.

**Figure 4 nanomaterials-12-01316-f004:**
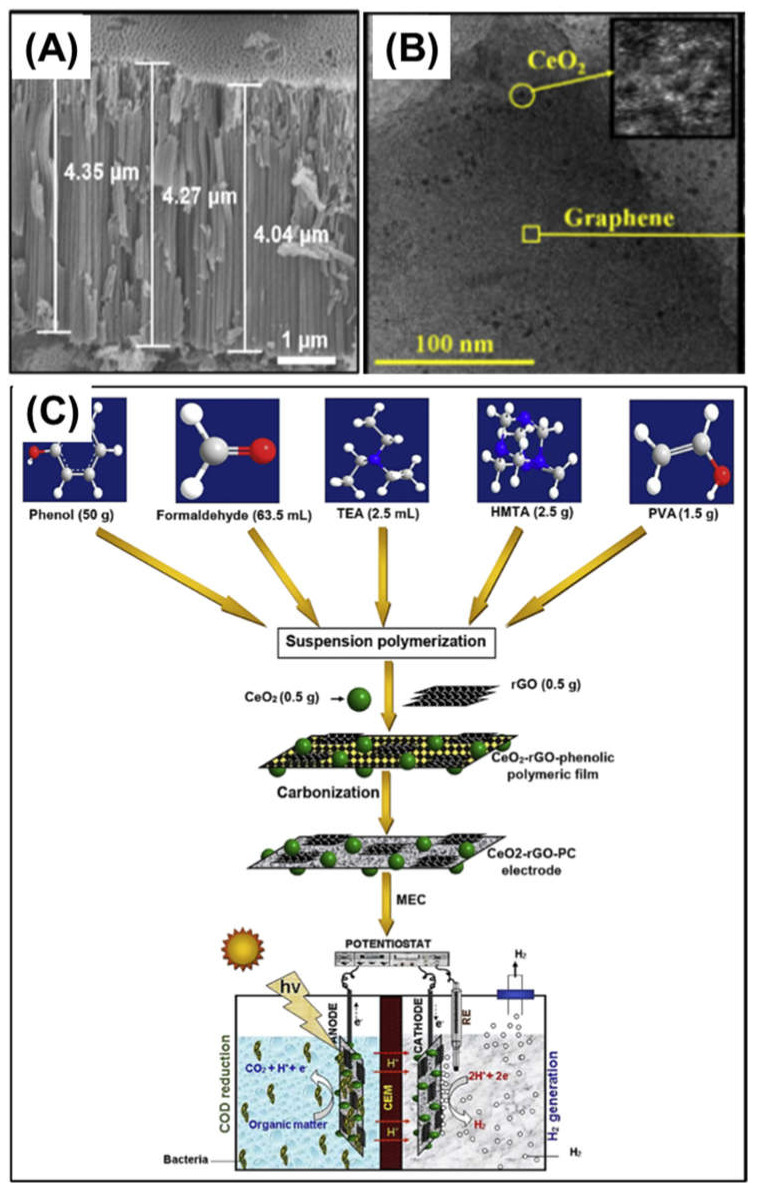
Nanomaterials applied to anode electrodes of MECs: (**A**) photocatalytic TiO_2_ nanotube array anode electrode. Reprinted with permission from Ref. [27]. Copyright MDPI 2018, (**B**) CeO_2_ nanoparticle-decorated rGO nanosheets. Reprinted with permission from Ref. [38]. Copyright Elsevier 2020, and (**C**) fabrication procedure for CeO_2_–rGO-incorporated carbon film anode electrode. Reprinted with permission from Ref. [38]. Copyright Elsevier 2020.

**Figure 5 nanomaterials-12-01316-f005:**
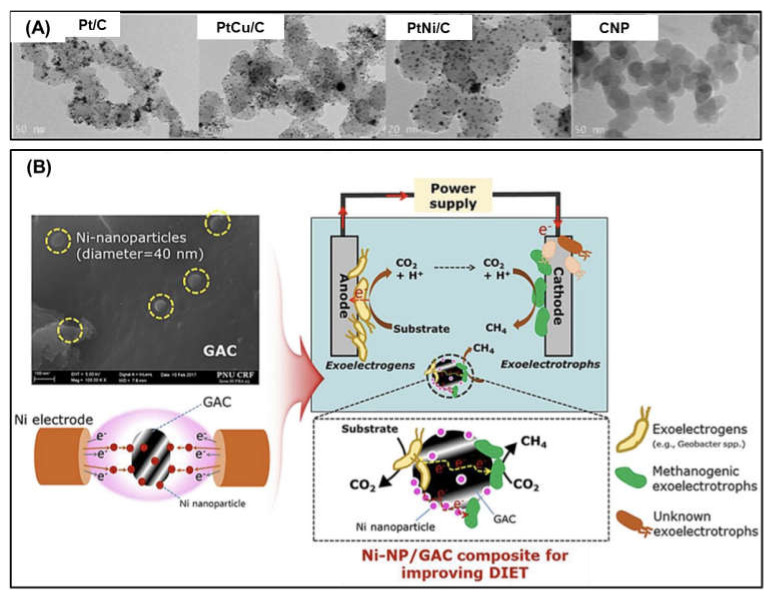
Metal nanoparticles used as cathode catalysts in microbial electrolysis cells: (**A**) metal nanoparticles decorated on carbon nanoparticles (CNPs). Reprinted with permission from Ref. [39]. Copyright Elsevier 2019. and (**B**) Granular activated carbon (GAC) electrodes decorated with Ni nanoparticles Reprinted with permission from Ref. [40]. Copyright Elsevier 2017.

**Figure 6 nanomaterials-12-01316-f006:**
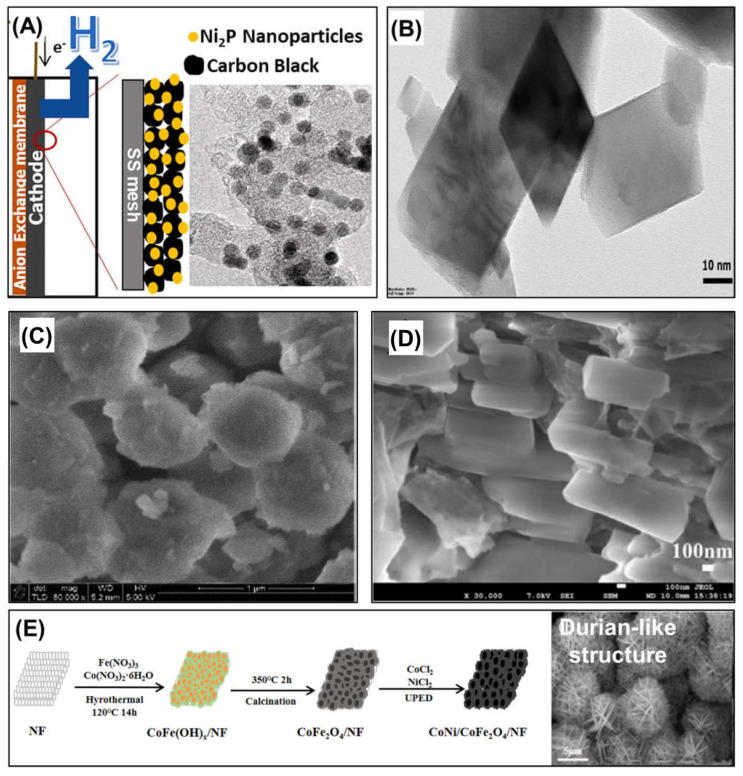
Metal compound-based nanomaterials employed as cathodic catalysts for hydrogen production: (**A**) Ni_2_P NPs on carbon black used as cathodic catalyst. Reprinted with permission from Ref. [42]. Copyright Elsevier 2019, (**B**) NiMoO_4_ NP catalysts for modifying a Ni foam cathodic electrode in an MEC. Reprinted with permission from Ref. [34] Copyright Elsevier 2020, (**C**) NiO-loaded Y zeolite nanocomposites. Reprinted with permission from Ref. [58]. Copyright Taylor & Francis 2019, (**D**) NiO/MoO_2_/MoO_3_/C nanocomposites. Reprinted with permission from Ref. [32] Copyright ESG 2019, and (**E**) synthesis procedure of CoNi/CoFe_2_O_4_ composite and a microscopic image of CoNi/CoFe_2_O_4_ composites with durian-like structure. Reprinted with permission from Ref. [33]. Copyright Wiley 2022.

**Figure 7 nanomaterials-12-01316-f007:**
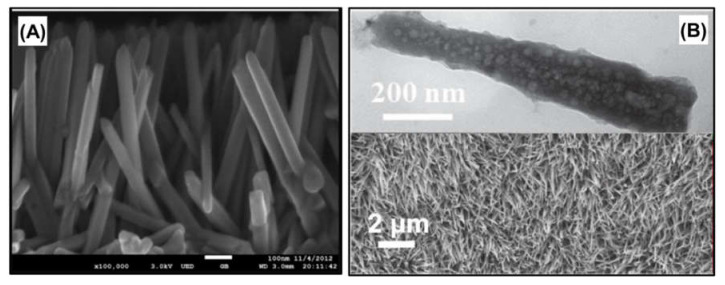
Transition metal-based nanomaterials employed as photocathodes in microbial electrolysis cells: (**A**) TiO_2_ nanorods. Reprinted with permission from Ref. [43]. Copyright Elsevier 2013. and (**B**) CoP nanoarrays. Reprinted with permission from Ref. [46]. Copyright Elsevier 2020.

**Figure 8 nanomaterials-12-01316-f008:**
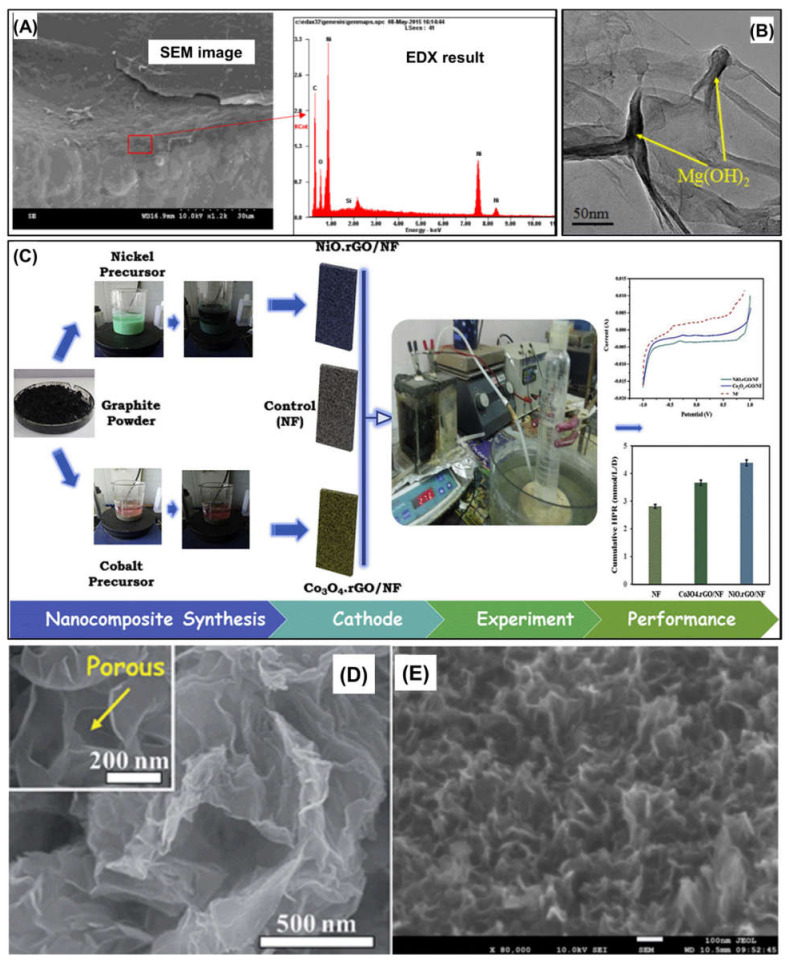
Two-dimensional nanomaterials used in MECs: (**A**) SEM image of graphene-coated Ni foam surface. Reprinted with permission from Ref. [50]. Copyright Elsevier 2016, (**B**) Mg(OH)_2_-decorated GO nanosheet. Reprinted with permission from Ref. [51]. Copyright Elsevier 2016, (**C**) schematic of NiO–rGO and Co_3_O_4_–rGO nanocomposites fabrication and their application in an MEC. Reprinted with permission from Ref. [52]. Copyright Elsevier 2020, (**D**) macroscopic and microscopic image of MoS_2_/N-doped GO composite electrode. Reprinted with permission from Ref. [55]. Copyright Royal Society of Chemistry 2014, and (**E**) vertically-created MoS_2_ on rGO. Reprinted with permission from Ref. [57]. Copyright ESG 2021.

**Figure 9 nanomaterials-12-01316-f009:**
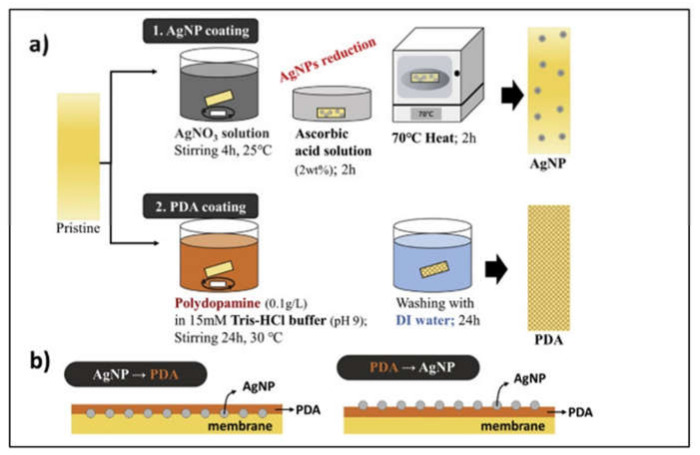
Membrane modification using silver nanoparticles (AgNPs) and polydopamine (PDA). (**a**) Procedures of AgNPs coating and PDA coating on membranes, and (**b**) Schematic comparing PDA coating after AgNP coating and AgNP coating after PDA coating. Reprinted with permission from Ref. [60]. Copyright Elsevier 2021.

## Data Availability

No new data were created or analyzed in this study. Data sharing is not applicable to this article.

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
