# Peer review of "Recent Application of Nanomaterials to Overcome Technological Challenges of Microbial Electrolysis Cells"

_nanomaterials, 2022, doi:10.3390/nano12081316_

Round 1

Reviewer 1 Report

The authors have made a remarkable review of the current state of studies for making the microbial electrolysis cells commercially attractive. The main focus is on the use of nanomaterials in different parts of the electrolysis cell. Extensive classification of nanomaterials is presented, each type is discussed. I recommend the manuscript for publication in Nanomaterials in present form. 

Author Response

Referee #1

Comment 1:  The authors have made a remarkable review of the current state of studies for making the microbial electrolysis cells commercially attractive. The main focus is on the use of nanomaterials in different parts of the electrolysis cell. Extensive classification of nanomaterials is presented, each type is discussed. I recommend the manuscript for publication in Nanomaterials in present form.

Our response) We thank the referee for providing us a positive review comment. 

Reviewer 2 Report

Referee’s report on paper  “Recent Application of Nanomaterials to Overcome Technological Challenges of Microbial Electrolysis Cells”, by Euntae Yang et al., Ms# 1628839

The manuscript reports on an interesting review of the contribution that nanomaterial research gives, or can give, to the development of Microbial Electrolysis Cells (MEC).

In my opinion the review can be of interest for the readers of “nanomaterials” since the developments and the state of the art described in the section 4 is comprehensive and stimulating. It is very interesting to be aware of all the ways nanomaterials could improve the performances of MEC. As the authors also realize, it is not clear to me whether scaling up (for industrial, large scale applications) will not present some unpredictable difficulties and bottlenecks, however, the perspectives of operating hydrogen harvesting by moderate power supplies (eolic or solar)  is stimulating and it is wise to spread knowledge about helpful nanomaterials for this purpose around.

The article needs improvements for the presentation especially as far as figures are concerned. In Fig. 1 some of the smallest characters are essentially unreadable and this isn’t really a good condition since this figure explains the working principles of MEC. Figure 2 is fine, for example. Figure 5 instead needs to be improved, some of it seem to be “out of focus” (especially (B)). Also, I note that there are two Figure 8 in the manuscript, one at page 16 and another at page 19. Thus, the order of the figures must me rearranged. I think the figure at page 19 must be Fig. 9. The lack of focus can occurn when figures, or parts of those  are taken from already existing/published material. I hope the authors are aware that if this is the case permission is needed to present those in a review work.

At line 82 (end of the line)  the reference must be specified.

I recommend the authors to carefully proof-read the paper.

Author Response

Referee #2

Comment 1:  The manuscript reports on an interesting review of the contribution that nanomaterial research gives, or can give, to the development of Microbial Electrolysis Cells (MEC). In my opinion the review can be of interest for the readers of “nanomaterials” since the developments and the state of the art described in the section 4 is comprehensive and stimulating. It is very interesting to be aware of all the ways nanomaterials could improve the performances of MEC. As the authors also realize, it is not clear to me whether scaling up (for industrial, large-scale applications) will not present some unpredictable difficulties and bottlenecks, however, the perspectives of operating hydrogen harvesting by moderate power supplies (eolic or solar) is stimulating and it is wise to spread knowledge about helpful nanomaterials for this purpose around.

Our response) We are grateful to the referee for reviewing our manuscript. We have carefully responded to and addressed the comments from the referee. As the reviewer’s suggestion,

Comment 2:  The article needs improvements for the presentation especially as far as figures are concerned. In Fig. 1 some of the smallest characters are essentially unreadable and this isn’t really a good condition since this figure explains the working principles of MEC.

Our response) As the reviewer’s comment, we resize the letters in Figure 1 to improve the readability.

Comment 3: Figure 5 instead needs to be improved, some of it seem to be “out of focus” (especially (B)). Also, I note that there are two Figure 8 in the manuscript, one at page 16 and another at page 19. Thus, the order of the figures must be rearranged. I think the figure at page 19 must be Fig. 9. The lack of focus can occurn when figures, or parts of those  are taken from already existing/published material. I hope the authors are aware that if this is the case permission is needed to present those in a review work.

Our response) We have changed some unfocused figures into high-resolution version. Also, we have already received copyright permissions of all the figures we used in this manuscript from publishers before we submitted. we have also revised the wrong designated figure numbers from Figure 8 to Figure 9 (on page 19) and from Figure 9 to Figure 10 (on page 21), respectively.

Comment 4: At line 82 (end of the line) the reference must be specified.

Our response) We have cited a reference for the sentence that the reviewer pointed out.

 ‘MECs are biocatalyst-assisted electrolysis devices for hydrogen generation [5].’

  1. Kadier, A.; Kalil, M.S.; Abdeshahian, P.; Chandrasekhar, K.; Mohamed, A.; Azman, N.F.; Logroño, W.; Simayi, Y.; Hamid, A.A. Recent advances and emerging challenges in microbial electrolysis cells (MECs) for microbial production of hydrogen and value-added chemicals. Renew. Sust. Energy Rev. 2016, 61, 501-525.

Comment 5: I recommend the authors to carefully proof-read the paper.

Our response) We are grateful to the referee for the comment. We have used a professional English editing service. In addition, we have carefully proofread our manuscript as the reviewer suggested.

Reviewer 3 Report

The manuscript contains a simple enumeration of the materials used without comparing their properties. Such representation of the data is absolutely useless.

I recommend to use scientific style without emotions. From this point of view, “The commercialization of MECs for hydrogen production is currently in a deadlock owing to major bottlenecks” should be changed.

No need to enter the designations of chemical elements: “platinum (Pt), nickel (Ni), palladium (Pd), copper (Cu), and gold (Au)”.

It is very likely that all the figures are taken from lectures for students and are not suitable for a scientific article.

Figure 4 as well as paragraph 3.1. Classification of nanomaterials should be removed because this is well-known information.

Author Response

Referee #3

Comment 1:  The manuscript contains a simple enumeration of the materials used without comparing their properties. Such representation of the data is absolutely useless.

Our response) First, we are grateful to the referee for reviewing our manuscript. However, not only do we disagree with some of your comments, but we are also severely displeased with the words you used. This is not a constructive review comment, but rather a personal rebuke.

Comment 2:  I recommend to use scientific style without emotions. From this point of view, “The commercialization of MECs for hydrogen production is currently in a deadlock owing to major bottlenecks” should be changed.

Our response) As the review suggested, we have revised the sentence into a scientific style.

Line 107, Page 3

“The commercialization of MECs for hydrogen production currently remains a stagnant state owing to significant challenges”

Comment 3:  No need to enter the designations of chemical elements: “platinum (Pt), nickel (Ni), palladium (Pd), copper (Cu), and gold (Au)”.

Our response) As the reviewer pointed out, we have eliminated symbols of chemical elements.

Comment 4:  It is very likely that all the figures are taken from lectures for students and are not suitable for a scientific article.

Our response) We do not agree with this comment. Most of the figures used in our manuscript, we adopted from relevant previous studies with copyright permissions from publishers. In addition, we have published many papers with the similar style of figures.

Comment 5:  Figure 4 as well as paragraph 3.1. Classification of nanomaterials should be removed because this is well-known information.

Our response) As the reviewer comment, we have removed Figure 4 and section 3.1.  

Round 2

Reviewer 3 Report

Unfortunately, there is no analysys of the reviewed data (Table 3). It is necessary to make conclusions about how the nanomaterial used affect performance. Which nanomaterial is better to use?

Author Response

Comment 1: Unfortunately, there is no analysys of the reviewed data (Table 3). It is necessary to make conclusions about how the nanomaterial used affect performance. Which nanomaterial is better to use?

Response:  We agree with the reviewer to some extent. However, it is very difficult to conclude which nanomaterials are better than the others by directly comparing the efficiency and performance of nanomaterials used in microbial electrolysis cells (MECs) because of the following reasons. First, the operational conditions for MECs in previous studies are different. Various factors affect MEC performance, such as reactor design, temperature, reactor volume, electrode size, substrate, etc.  Therefore, it is not easy to normalize the efficiency of nanomaterials based on the performance of MECs. Second. as we mentioned in our manuscript, the application purpose of each nanomaterial is different,  such as cathodic catalyst, membrane reinforcement, and electrical conductivity improvement of anode electrodes.